# Intracellular *Cryptococcus neoformans* disrupts the transcriptome profile of M1- and M2-polarized host macrophages

Aarthi Subramani[1], Prianca Griggs[1], Niah Frantzen[1], James Mendez[1],
Jamila Tucker[1,2], Jada Murriel[1], Linda M. Sircy[1,3], Grace E. Millican[1], Erin
E. McClelland[1,4], Rebecca L. Seipelt-Thiemann[1], David E. Nelson[1]*

**1** Biology Department, Middle Tennessee State University, Murfreesboro, TN, United States of America,
**2** Microbiology, Immunology, and Molecular Genetics Department, University of Kentucky, Lexington, KY,
United States of America, **3** Department of Pathology, University of Utah, Salt Lake City, UT, United States of
America, **4** M&P Associates, Inc., Murfreesboro, TN, United States of America

* david.e.nelson@mtsu.edu

doi.org/10.1371/journal.pone.0233818

Health System, UNITED STATES

**Data Availability Statement:** All relevant data are
within the manuscript and its Supporting
Information files. Original RNA sequencing files
produced during this study are deposited in the

## Abstract

Macrophages serve as a first line of defense against infection with the facultative intracellular pathogen, *Cryptococcus neoformans* (*Cn*). However, the ability of these innate phagocytic cells to destroy ingested *Cn* is strongly influenced by polarization state with classically (M1) activated macrophages better able to control cryptococcal infections than alternatively (M2) activated cells. While earlier studies have demonstrated that intracellular *Cn* minimally affects the expression of M1 and M2 markers, the impact on the broader transcriptome associated with these states remains unclear. To investigate this, an *in vitro* cell culture model of intracellular infection together with RNA sequencing-based transcriptome profiling was used to measure the impact of *Cn* infection on gene expression in both polarization states. The gene expression profile of both M1 and M2 cells was extensively altered to become more like naive (M0) macrophages. Gene ontology analysis suggested that this involved changes in the activity of the Janus kinase-signal transducers and activators of transcription (JAK-STAT), p53, and nuclear factor-κB (NF-κB) pathways. Analyses of the principle polarization markers at the protein-level also revealed discrepancies between the RNA- and protein-level responses. In contrast to earlier studies, intracellular *Cn* was found to increase protein levels of the M1 marker iNos. In addition, common gene expression changes were identified that occurred post-*Cn* infection, independent of polarization state. This included upregulation of the transcriptional co-regulator *Cited1*, which was also apparent at the protein level in M1-polarized macrophages. These changes constitute a transcriptional signature of macrophage *Cn* infection and provide new insights into how *Cn* impacts gene expression and the phenotype of host phagocytes.

## Introduction

The basidiomycetous yeast *Cryptococcus neoformans* (*Cn*) is a common facultative intracellular pathogen and the causative agent of cryptococcosis, a pulmonary infection that predominantly

NCBI GEO under the accession number
GSE150802.

**Funding:** This work was supported by funds from
the National Institutes of Health (NIAID
1R15AI135826-01) to DEN, EEM, and RLST and
the Molecular Biosciences (MOBI) doctoral
program at Middle Tennessee State University
(MTSU) to DEN, and AS. M&P Associates, Inc.
provided support in the form of salaries for authors
[EEM], but did not have any additional role in the
study design, data collection and analysis, decision
to publish, preparation of the manuscript, or
obtaining research materials. The specific roles of
this author are articulated in the 'author
contributions' section.

**Competing interests:** EEM was employed by M&P
Associates, Inc. after leaving MTSU. M&P
Associates, Inc. did not have any role in the study
design; collection, analysis, and interpretation of
data; writing of the paper; obtaining research
materials; and/or decision to submit for
publication. This does not alter our adherence to
PLOS ONE policies on sharing data and materials.
All other authors have declared that no competing
interests exist.

affects immune-compromised individuals and that can disseminate to the central nervous system, resulting in life-threatening fungal meningitis [1, 2]. Cryptococcal infections occur when propagules, typically from bird excreta-contaminated soils, are inhaled into the lungs. Here, they encounter alveolar macrophages, innate phagocytes that act as a first line of defense against the pathogen. This interaction between host macrophages and *Cn* is perhaps the most important in determining the course and outcome of an infection [3–5], and there is substantial evidence to show that macrophages are essential for the successful control of cryptococcosis [6, 7].

The ability of macrophages to efficiently kill ingested *Cn* is greatly influenced by the polarization state of these cells [8–14]. Macrophage polarization is a continuum of phenotypes of differing function and microbicidal activity requiring the altered expression of >1000 genes [15, 16]. These do not represent terminal differentiation states as macrophages can rapidly repolarize over the course of hours from one state to another in response to microbe and immune cell-derived signals (i.e. cytokines) [17]. Interferon-gamma (IFNγ) or lipopolysaccharide (LPS) stimulates classical activation or M1 polarization and is associated with proinflammatory cytokine expression and metabolic shifts that increase production of microbicidal reactive oxygen and reactive nitrogen species [18]. This is partially achieved through increased expression of the gene, *Nos2*, which encodes inducible nitric oxide synthase (iNOS) and catalyzes the production of nitric oxide (NO) from L-arginine [19]. As infections are resolved, elevated levels of interleukin-4 (IL-4) and IL-13 promote the repolarization of macrophages to an anti-inflammatory alternative or M2 polarization state, which is accompanied by increased expression of arginase-1 (Arg1) [20]. This enzyme competes with iNos for substrate, thereby reducing NO production and macrophage microbicidal activity [21].

Infection with the virulent H99S strain of *Cn* initially promotes a Th2-response in mice and the associated changes in cytokine production stimulate M2 macrophage polarization [10, 11, 20, 22]. This polarization state presents a less hostile environment for intracellular *Cn*, serving as a protective niche for growth and immune evasion (reviewed in [23]). Previous studies have suggested that *Cn* has only modest effects on the polarization state of host cells, minimally affecting the expression of *Nos2* and *Arg1*, the principle markers of the M1 and M2 states [17]. However, the impact on the broader transcriptome associated with these states remains unclear and is not captured in earlier microarray or RNA sequencing-based analyses of *Cn*-infected macrophages where polarization state is not explicitly considered [24–26].

Here, we present data showing that intracellular *Cn* infection resulted in extensive changes to the transcriptome of host macrophages, shifting the gene expression profiles of both M1 and M2 cells to a more naïve (M0) state, that is macrophages that have not previously been exposed to proinflammatory or anti-inflammatory cytokines, while causing relatively small changes in the expression of *Nos2* and *Arg1*. Additionally, a transcriptional signature of *Cn* infection, common to both polarization states was identified, which included upregulation of the transcriptional co-regulator *Cited1*. Collectively, these results provide new insights into how *Cn* reshapes gene expression and the phenotype of host cells.

## Materials and methods

### Culture and opsonization of *Cn*

The serotype A H99S strain [27] of *Cn* serotype A was grown in yeast peptone dextrose (YPD; ThermoFisher Scientific) broth, shaking at 37°C for 36 h prior to infection. After 36 h, $1 \times 10^7$ *Cn* cells were washed 3× with phosphate-buffered saline (PBS), pelleted by centrifugation at $750 \times g$ for 5 min, and then counted. The cells were opsonized with 18B7 (a kind gift from Dr. Arturo Casadevall; previously described in [28]) using 20 μg per $1.5 \times 10^6$ cells in 1 mL 20%

goat serum (GS; Sigma Aldrich, St. Louis, MO) for 30 min. The opsonized cells were washed 3× with PBS to remove excess 18B7 and were recounted. A total of 2.25 x 10$^6$ cells/mL were necessary to ensure a 3:1 multiplicity of infection (MOI; *Cn*:macrophage).

## Macrophage culture, polarization, and infection

RAW 264.7 cells, a murine macrophage-like cell line, were obtained from the American Type Culture Collection (ATCC, Manassas, VA) and cultured in Dulbecco's Modified Eagle Medium (DMEM; ThermoFisher Scientific, Grand Island, NY) supplemented with 10% fetal bovine serum (FBS; VWR, Radnor, PA), 200 mM L-glutamine, 1% penicillin and streptomycin, and 50 μg/mL gentamicin (all from Sigma Aldrich). Macrophages were maintained at 37˚C in a humidified 5% CO$_2$ atmosphere.

Prior to infection, RAW 264.7 cells were seeded into 6-well plates at a density of 7.5 x 10$^5$ cells/well and incubated overnight with 200 U/mL of recombinant murine IFNγ (Biolegend, San Diego, CA) to promote M1-polarization and then infected with opsonized *Cn* for 2 h at a 3:1 MOI or mock infected with PBS. After 2 h, extracellular *Cn* were removed by washing 2× with PBS, and the macrophages were cultured for a further 24 h in fresh growth media containing either IFNγ to maintain the M1 polarization state or 100 ng/mL recombinant interleukin-4 (IL-4; Sigma-Aldrich, St. Louis, MO, and Peprotech, Rocky Hill, NJ) to promote repolarization to the M2 state. The polarization state was confirmed by western blotting for the M1 and M2 markers, iNos and Arg-1, respectively. Cells were washed and growth medium containing IFNγ or IL-4 was replaced every 6 h to prevent nutrient depletion, remove extracellular *Cn*, and maintain the appropriate polarizing environment.

## Quantification of infection efficiency

RAW264.7 cells were seeded into 35 mm glassbottom dishes at a density of 7.5 x 10$^5$ cells/dish and incubated overnight with 200 U/mL of recombinant murine IFNγ and/or 1μg/mL of LPS to promote M1-polarization. The cells were infected with GFP-expressing H99S opsonized using GS and/or 18B7 anti-GXM antibodies (1× = 10 μg and 2× = 20 μg 18B7 per 1.5×10$^6$ *Cn*) at the indicated concentrations. The cells were cultured for a further 24 h in fresh growth media containing either IFNγ to maintain the M1 polarization state or IL-4 to promote repolarization to the M2 state. Dead macrophages were labeled with propidium iodide and extracellular *Cn* were labeled with calcofluor white (Sigma Aldrich, St. Louis, MO). The samples were imaged used a Zeiss LSM700 confocal laser scanning microscope equipped with a Plan-Apochromat 20×/0.8 M27 objective (Carl Zeiss). The percentage of live macrophages containing intracellular *Cn* was determined for each sample.

## Glucose assays

RAW264.7 cells were grown in phenol red-free DMEM adjusted to contain the indicated concentration of glucose at the start of the experiment. Growth medium from the cultures was sampled (1 mL) at the indicated times post-treatment and centrifuged to remove *Cn* and cellular debris. Glucose concentration was measured using a glucose oxidase assay kit (GAGO20, Sigma Aldrich) according to manufacturer's instructions.

## Immunoblotting

Macrophages were lysed using radioimmunoprecipitation assay (RIPA) buffer containing 1x protease inhibitor cocktail (Sigma-Aldrich) and 1 mM phenylmethylsulfonyl fluoride (PMSF, Sigma-Aldrich). Harvested samples were vortexed and centrifuged at 13,000 × *g* for 15 min at

4˚C to remove cell debris. Sample protein concentrations were measured using a bicinchoninic acid (BCA) assay kit according to manufacturer's instructions. Normalized protein lysates were boiled at 95˚C for 10 min in Laemmli sample buffer, electrophoresed on 10 or 12% SDS-polyacrylamide gels, and transferred to nitrocellulose membranes. The membranes were blocked in Tris-buffered saline/Tween 20 (TBS/T) containing 5% non-fat milk powder for 1 h and incubated overnight at 4˚C with the appropriate primary antibodies. These included β-actin (A2066, Sigma), iNos (D6B6S, Cell Signaling Technology, Danvers, MA), Arg1 (ab124917, Abcam, Cambridge, MA), MSG1 (Cited1; sc-393585, Santa Cruz Biotechnology, Dallas, TX), and VDUP1 (TXNIP; sc-271238, Santa Cruz Biotechnology). Primary antibody binding was detected using mouse anti-rabbit IgG-horseradish peroxidase (HRP; sc-2357, Santa Cruz Biotechnology) or anti-mouse m-IgGkappa binding protein (BP)-HRP (sc-516102, Santa Cruz Biotechnology), as appropriate. Membranes were incubated with enhanced chemiluminescent (ECL) reagents and bands were visualized using a ChemiDoc MP Imaging System with Image Lab Software (Bio-Rad, Hercules, CA).

## RNA extraction and cDNA library preparation

Immediately prior to harvest, all RAW264.7 samples were imaged by light microscopy to determine the percentage of cells infected with intracellular *Cn*. With the exception of the mock-infected samples, *Cn*-infected macrophage samples were only processed further if >50% of cells were infected. Macrophages were homogenized in lysis reagent (RLT buffer, Qiagen, Germantown, MD) using a 20-gauge needle and the remaining intact *Cn* were removed by centrifugation at 750 x *g* for 5 min at 4˚C. RNA was extracted from the lysates using an RNeasy® Mini Kit (Qiagen), according to the manufacturer's instructions. Genomic DNA was then removed from the total RNA using a Message Clean kit (GenHunter, Nashville, TN) per the manufacturer's instructions. Clean RNA was resuspended in 10 μL diethyl pyrocarbonate (DEPC)-treated water and RNA integrity and quality was appraised using a Qubit 2 fluorometer (ThermoFisher Scientific).

The cDNA libraries for RNA sequencing were prepared from 1 μg of isolated RNA using the NEBNext UltraTM Directional RNA Library Prep Kit for Illumina, the NEBNext Multiplex Oligos for Illumina Index Primers, and the NEBNext Poly(A) Magnetic Isolation Module (all from New England BioLabs, Ipswich, MA) in accordance with the manufacturer's instructions.

## Analysis of RNA sequencing data

RNA sequencing of each library sample was performed at Novogene (Sacramento, CA) using the HiSeq 2500 system to produce 150 bp transcriptome paired-end reads. FastQC (version 0.11.5; [29]) was used to check quality of the fastq data files. No trimming of the files was necessary based on the data quality. The STAR aligner (version 2.5.3a; [30]) was used to align reads to the version 38 mouse genome [31] with scaffolding provided by the mouse reference genome annotation (version 39.90, [31]) within the CyVerse Discovery Environment [32]. The resulting bam and the same mouse genome annotation were used to generate a read count table by gene using FeatureCounts [33] and multi-join [34] within the Galaxy platform [35]. These read counts were imported into R where they were used to cluster samples according to their whole genome gene expression profile using EdgeR [36], as detailed in [37]. Data were displayed as both a multi-dimensional scale plot and dendrogram to aid in evaluating samples for inclusion or exclusion. Three samples per condition (a total of 15) were initially analyzed. Of these, four samples were excluded from further analysis due to their failure to cluster within their replicate pool (M1_mock replicate 2, M1_Cn replicate 1, M2_mock replicate 3, and M0

replicate 1). Next, StringTie (version 1.3.3; [38]) was used to construct transcript annotations based on the bam data and the same mouse reference genome and genome annotation within the CyVerse Discovery Environment [32]. StringTieMerge (version 1.3.3;[38]) was then used to merge these sample-specific genome annotations into an experiment-specific genome annotation, also within the CyVerse Discovery Environment [32]. Finally, CuffDiff2 (version 2.2.1; [39]) was then used to make pairwise comparisons for differentially expressed genes (DEGs) among the replicate groups of aligned reads (bam files) based on the experiment-specific genome annotation within the Cyverse Discovery Environment [32]. From the pairwise comparisons, DEGs with fold change $\geq$ 2.0 and q $\leq$ 0.05 were considered biologically relevant and statistically significant. Functional categorization and pathway over-representation analysis of genes within the statistically significant DEGs were performed using the Database for Annotation, Visualization, and Integrated Discovery (DAVID) bioinformatics resource tool (version 6.8; [40, 41]) to identify biological pathways specific to polarization state and infection state. Gene ontology (GO) terms were ranked by *p*-value, which was plotted as–log(*p*-value).

Finally, protein-protein interaction networks were identified among DEG sets using the Search Tool for the Retrieval of Interacting Genes/Proteins (STRING, version: 11.0; [42]). The confidence interval was set at 0.700. Evidence based analysis was used with Markov Cluster (MCL) grouping. Common protein-protein interaction clusters were then visually identified for M0-M1mk compared to M0-M1*Cn*, M0-M2mk compared to M0-M2*Cn*, M0-M1*Cn* compared to M0-M2*Cn*, and M1-M1*Cn* compared to M2-M2*Cn*. Protein network changes based on polarization state and infection status were identified by determining the common and unique genes/proteins between the common clusters for each pair.

## Results

### Development of a strategy to obtain *Cn*-infected M1 and M2-polarized macrophages

To develop an *in vitro* system to accurately characterize the effects of intracellular *Cn* infection on the transcriptome of host macrophages in different polarization states, it was necessary to i) utilize a macrophage cell line that can adopt clear M1 and M2 phenotypes, ii) maintain a stable growth environment for the cells so that nutrient depletion did not impact gene expression, and iii) be able to infect these cells with high efficiency so that the vast majority of mRNA transcripts identified in the RNA sequencing analysis originated from *Cn*-infected cells.

For this study, the murine macrophage-like cell line RAW 264.7 was selected, which is frequently used as an *in vitro* model to investigate how intracellular pathogens affect macrophage polarization and gene expression [43, 44] and how gene expression of pathogens is altered in host cell environments [45]. It has also been used by us and others to study macrophage:*Cn* interactions [17, 46].

While naive RAW 264.7 cells can be stimulated to adopt an M1-like phenotype through exposure to IFNγ alone, expressing various proinflammatory cytokines and detectable levels of the M1 marker, iNos [17], it is unclear whether these cells can be directly polarized to the M2 state. Stimulation of naive RAW 264.7 cells with IL-4 has been shown to induce transcriptional upregulation of the M2 marker, *Arg1*, but does not result in detectable levels of the Arg-1 protein [47]. This was also true in our hands and we were only able to stimulate a measurable increase in Arg-1 protein expression if RAW 264.7 cells were co-stimulated with IL-4 and the cAMP analog, 8-bromo-cAMP to activate C/EBPbeta, a co-regulator of the *Arg1* promoter [47] (Fig 1A).

As macrophages will typically repolarize from an M1 to an M2 state rather than directly from M0 to M2 during the course of a normal infection, we reasoned that RAW 264.7 macrophages may operate in a similar fashion. To test this, RAW 264.7 cells were stimulated with

IFNγ for 24 h followed by treatment with IFNγ or IL-4 for a further 24 h. As expected, continuous IFNγ treatment resulted in ~300 fold increase in iNos protein levels (Fig 1B and 1C). While Arg-1 protein levels were indistinguishable from untreated controls and samples that

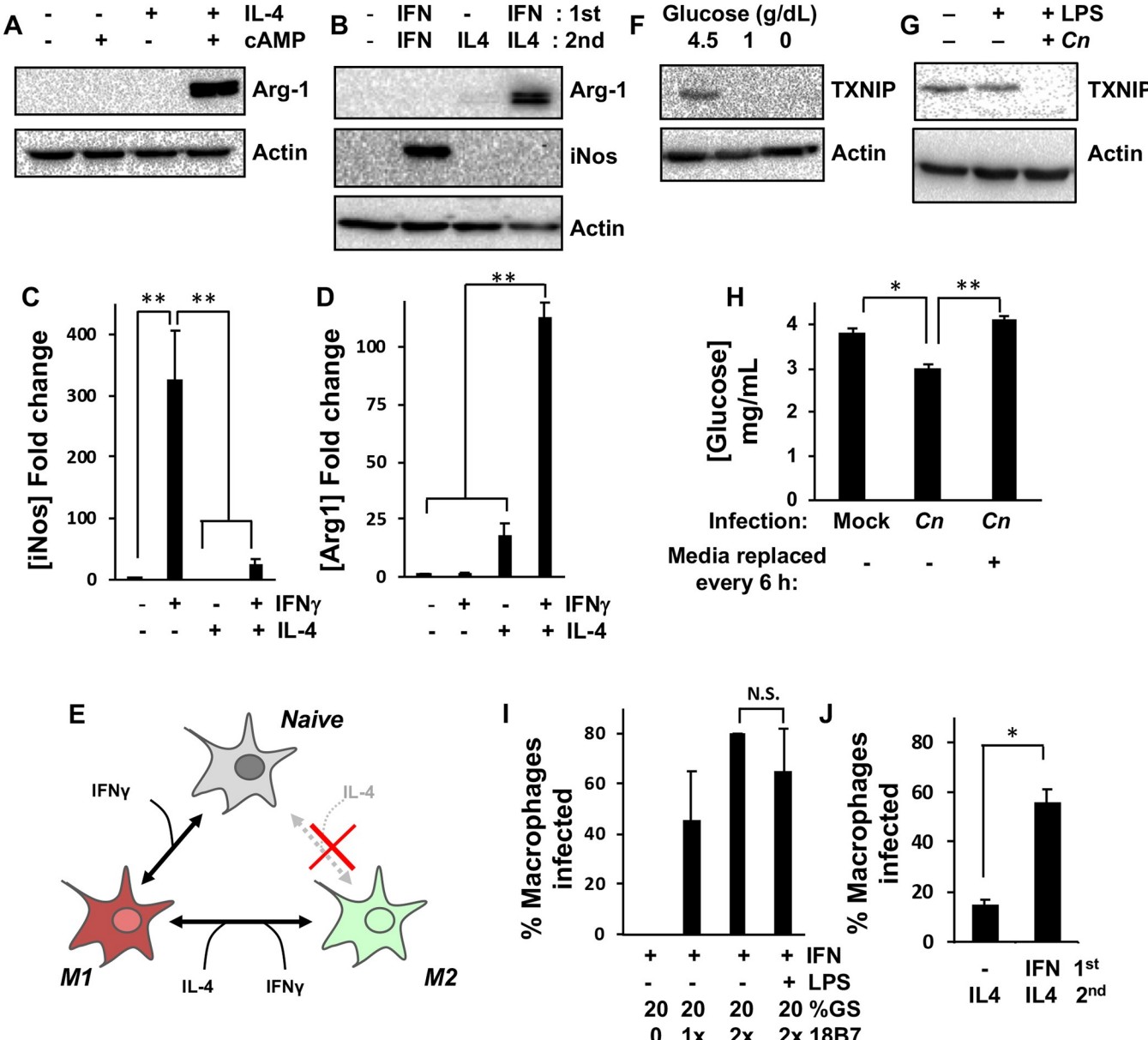

**Fig 1. Optimization of murine macrophage intracellular *Cn* infection for transcriptome profiling.** Western blot analysis of RAW264.7 macrophages for M1 (iNos) and/or M2 (Arg-1) marker proteins after incubation for 24 h with **(A)** IL-4 and cAMP or **(B)** IFNγ and IL-4. **(C)** iNos and **(D)** Arg-1 levels in *(B)* were quantified by densitometry based on six discrete biological repeats. **(E)** Schematic to summarize data from *(A-D)*, which suggests that while RAW264.7 cells cannot be directly polarized from M0 to M2 by IL-4 treatment, they can be repolarized from M1 to M2. Expression of TXNIP, a marker of glucose levels, as measured in RAW264.7 macrophages by western blotting after 24 h **(F)** incubation in growth medium containing the indicated glucose concentrations or **(G)** infection with 18B7-opsonized *Cn*. LPS was added for 2 h during infection to promote phagocytosis of *Cn*. **(H)** Glucose concentration as measured in RAW264.7 macrophage growth medium using a glucose oxidase assay kit 24 h post-mock or *Cn*-infection. Complete replacement of growth medium at 6 h intervals to remove extruded *Cn* prevents glucose depletion. **(I and J)** The percentage of *Cn*-infected RAW264.7 cells was quantified by confocal microscopy 24 h post-infection. For all experiments, concentrations are as follows: 200 U/mL IFNγ, 100 ng/mL IL-4, 1 µg/mL LPS, cAMP is 0.5 mM 8-Br-cAMP, 1× 18B7 is 10 µg per $1.5 \times 10^6$ *Cn*, GS is goat serum. Error is represented as S.E. Statistical differences between samples were appraised using one-way analysis of variance (ANOVA) followed by a Tukey's multiple comparison test. Statistical significance is indicated as follows: *, $p < 0.05$; **, $p < 0.01$. With the exception of *(B-D)*, data is from three biological repeats.

did not receive IFNγ prior to IL-4 stimulation, repolarization from M1 to M2 resulted in >100 fold increase in Arg-1 protein levels (Fig 1D). Collectively, these data suggested that RAW 264.7 cells could be used to model both the M1 and M2 state with the caveat that the M2 phenotype could not be reached directly from the naive state but via repolarization from M1 (Fig 1E).

Given that *Cn* replicates within host cell phagolysosomes and escapes into the surrounding culture medium through non-lytic exocytosis or vomocytosis [48–51], we reasoned that the accumulation of live, extracellular *Cn* might result in the accelerated depletion of glucose and other nutrients from the culture medium. As this could potentially result in confounding changes in gene expression not directly caused by intracellular *Cn* growth, the expression of the glucose-regulated gene, thioredoxin–interacting protein (*TXNIP*) was used as a marker of glucose depletion-induced gene expression changes in these experiments. TXNIP is a regulator of nitrosative stress and glucose metabolism in a variety of cell types and is itself regulated at a transcriptional level by heterodimers of the glucose-responsive transcription factors, MondoA and Mlx [52–54]. Decreasing glucose levels from 4.5 g/dL in the macrophage culture medium to 1.0 g/dL for 24 h resulted in a complete loss of TXNIP protein expression in RAW 264.7 cells, indicating that TXNIP expression was a suitable marker of glucose depletion in this system (Fig 1F).

In preliminary RNA sequencing experiments (unpublished), the expression of *TXNIP* transcripts was decreased 8.34 fold ($q$ = 0.0016) in *Cn* versus mock-infected RAW 264.7. Additionally, in parallel immunoblotting experiments, it was found that the TXNIP protein dropped to undetectable levels at 24 h post-*Cn* infection (Fig 1G). Although the decrease in glucose concentration in *Cn*-infected cultures at 24 h post-infection was relatively modest (~20%; Fig 1H), these data suggested that the loss of *TXNIP* expression was artefactual and that *Cn*-induced changes in culture conditions over the course of the 24 h experiments may impact the resultant transcriptome profiles. In an effort to mitigate this in all subsequent experiments, macrophage cultures were washed, and the medium was replaced at 6 h intervals to remove extracellular *Cn* and prevent large changes in culture conditions (Fig 1H).

As a standard RNA sequencing approach was utilized, harvesting RNA from large populations of cells, phagocytosis conditions were optimized to maximize the number of *Cn*-infected M1 and M2-polarized macrophages in the cultures. This was achieved by titrating the concentration of 18B7 [28], a monoclonal antibody raised against *Cn* capsular polysaccharide that is used to opsonize the yeast prior to infection of M1-polarized macrophages (Fig 1I). It was found that a concentration of 2× 18B7 (20 μg for 1.5 x 10$^6$ cells) was sufficient to routinely obtain infection efficiencies of ~80%, as determined at the time of macrophage harvest for mRNA extraction. While it is common to add LPS to macrophages to promote phagocytosis, it was found that this did not affect the percentage of *Cn*-infected macrophages (Fig 1I). Furthermore, as co-infections with gram-negative bacteria are uncommon in cryptococcosis patients [55], LPS was excluded from all subsequent transcriptome-profiling experiments. Finally, it was found that infection efficiencies of >50% could be achieved for M2 macrophages if the cells were IFNγ-stimulated 24 h prior to infection and then repolarized to M2. If the cells were not M1 polarized first, not only was Arg-1 protein expression lost (Fig 1B–1D), but also the percentage of *Cn*-infected macrophages decreased to <20% (Fig 1J).

## Intracellular *Cn* increased iNos protein expression in M1-polarized macrophages

Based on the data described in Fig 1, a strategy was developed to produce *Cn*-infected M1 and M2-polarized RAW 264.7 cells (Fig 2A). In brief, this involved M1-polarizing naive

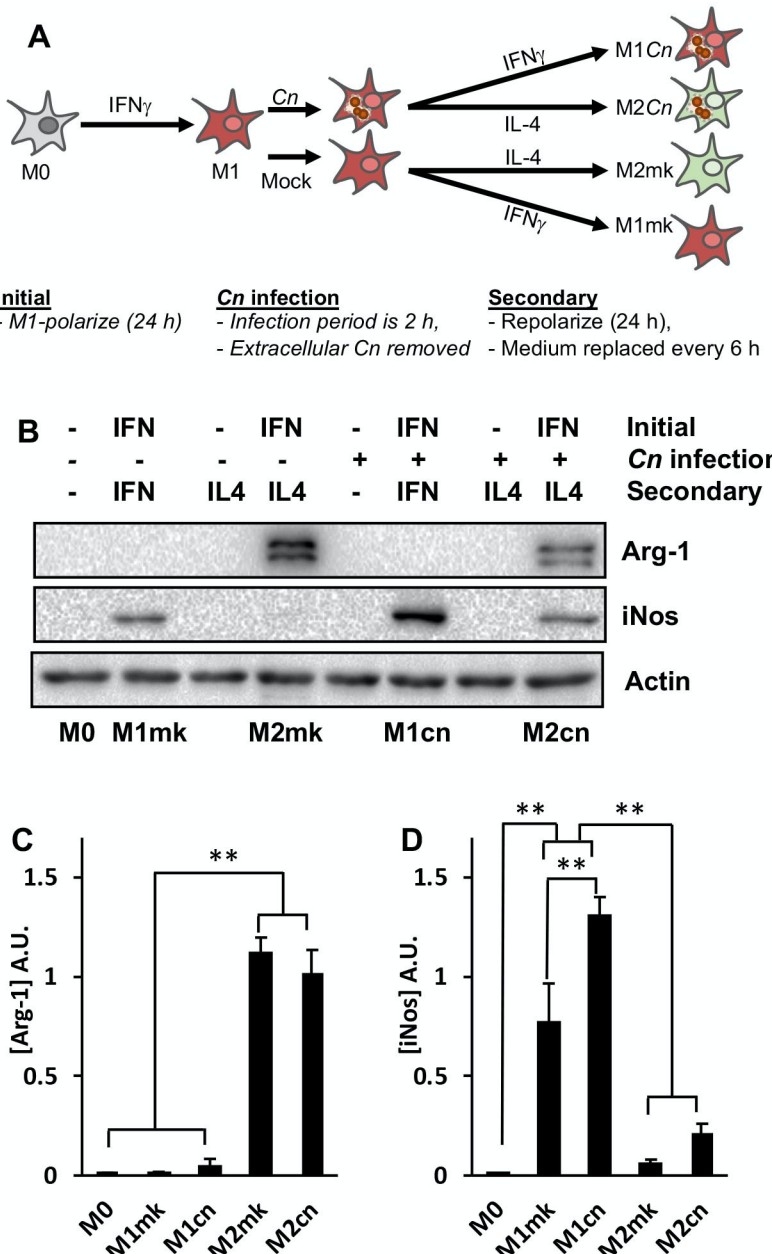

**Fig 2. Expression of iNos is increased in *Cn*-infected M1-polarized macrophages. (A)** Schematic to represent macrophage infection protocol used for subsequent western blotting and transcriptome profiling experiments. **(B)** Western blot analysis of RAW264.7 macrophages for M1 (iNos) and M2 (Arg-1) markers after the indicated treatments. **(C)** iNos and **(D)** Arg-1 levels in *(B)* were quantified by densitometry based on six discrete biological repeats. Error is represented as S.E. Statistical differences between samples were appraised by one-way ANOVA followed by a Tukey's multiple comparison test. Statistical significance is indicated as follows: **, $p < 0.01$.

macrophages by incubation with IFNγ for 24 h, then mock or *Cn*-infection with complement- (provided by 20% GS) and 18B7-opsonized yeast at an MOI of 3:1. After a 2 h infection period, residual extracellular *Cn* were removed by washing and the cells were cultured for a further 24 h in growth medium containing either IFNγ to maintain the M1 polarization state or IL-4 to repolarize to M2, with the growth medium and cytokines replaced at 6 h intervals.

**Table 1. M0 vs. M1mk.**

| Gene | FC | Direction | *q*-value |
|------|-----|-----------|-----------|
| Cxcl9 | 682.63 | UP | 0.00626191 |
| Gbp2 | 369.19 | UP | 0.00626191 |
| Cd86 | 356.53 | UP | 0.0383977 |
| Nos2 | 150.18 | UP | 0.00626191 |
| Cxcl10 | 139.62 | UP | 0.00626191 |
| Stat1 | 46.87 | UP | 0.0334391 |
| IL1b | 46.41 | UP | 0.0161112 |
| Fcgr1 | 17.61 | UP | 0.00626191 |
| Ccl5 | 6.66 | UP | 0.0370904 |

Expression of M1 macrophage markers (FC = Fold-change)

This protocol was utilized to perform RNA sequencing-based transcriptome profiling of naive (M0), mock- (M1mk and M2mk) and *Cn*-infected (M1*Cn* and M2*Cn*) macrophages in both polarization states. As an initial quality control step, pairwise comparisons of gene expression in M0 with M1mk or M2mk samples was performed and changes in the expression of a small panel of known polarization markers were examined [15, 16, 56]. As expected, *Nos2*, *Stat1*, a range of M1-associated cytokines and chemokines (*Il1b*, *Cxcl9*, *Cxcl10*, and *Ccl5*), and surface markers (*Fcgr1* and *Cd86*) were strongly upregulated (Table 1). A similar analysis was performed for cells repolarized from M1 to M2, and in this case, upregulation of the M2 markers, *Arg1* and *IL10*, as well as downregulation of M1 markers was observed (Table 2).

Consistent with previous reports [17], *Cn* infection appeared to have little effect on the expression of these core polarization markers. The expression of *Nos2* was not significantly altered in M1- or M2-polarized cells during *Cn* infection, and amongst the remaining M1 markers, only *Ccl5* was increased in M1mk vs. M1*Cn* (FC = 3.71, *q* = 0.046). Surprisingly, *Arg1* was decreased in M2mk vs. M2*Cn* (FC = 3.97, *q* = 0.006). To determine whether these effects were also apparent at the protein level, Arg-1 and iNos levels were measured by western blotting. Surprisingly, while Arg-1 was seemingly unaffected by *Cn* infection in both polarization states (Fig 2B and 2C), suggesting additional regulatory controls, iNos levels were increased by ~40% in M1-polarized cells (Fig 2B–2D).

## Distortion of the M1 and M2 transcriptome in *Cn*-infected macrophages

Although the initial analysis suggested that *Cn* infection minimally affected principle polarization state markers, we could not rule out the possibility that the broader transcriptome and key processes associated with each state were impacted. To comprehensively examine this,

**Table 2. M1mk vs. M2mk.**

| Gene | FC | Direction | *q*-value |
|------|-----|-----------|-----------|
| Arg1 | 623.88 | UP | 0.00626191 |
| Atp6v0d2 | 14.06 | UP | 0.0350194 |
| IL10 | 10.08 | UP | 0.0413593 |
| Cxcl10 | 18.9 | DOWN | 0.00626191 |
| Cxcl11 | 15.77 | DOWN | 0.0200165 |
| Nos2 | 12.24 | DOWN | 0.00626191 |
| Cxcl9 | 8.84 | DOWN | 0.00626191 |
| GBP2 | 5.21 | DOWN | 0.00626191 |

differentially expressed genes (DEGs) from each state were identified through pairwise comparisons of M0 to M1mk or M2mk RNA sequencing data using a fold change of 2 as a cut-off. These were then compared to corresponding DEGs from M0 to M1*Cn* or M2*Cn* comparisons.

Of the 931 DEGs associated with M0 to M1 polarization, 332 (~36%) were common to the 460 DEGs from the M0:M1*Cn* comparison (Fig 3A). This suggested that the expression of the remaining 599 DEGs (~64%) associated with the M1 state became or remained more M0-like post-*Cn* infection. The 128 DEGs (~28%) unique to the M0:M1*Cn* comparison were assumed to be *Cn*-induced changes in gene expression not associated with the normal M1 transcriptional profile. Kyoto Encyclopedia of Genes and Genomes (KEGG) pathway analysis of the common pool of DEGs showed an enrichment of genes associated with core M1 processes, including 'JAK-STAT signaling', 'Cell adhesion molecules', and 'Cytosolic DNA-sensing pathway'. Interestingly, a subset of these terms, including 'Phagosome', 'Antigen processing and presentation', 'Toll-like receptor signaling', and 'Tumor necrosis factor (TNF) signaling pathway' also appeared significant for the M0:M1mk-exclusive DEG pool, suggesting changes to these M1-associated processes during *Cn* infection. Similarly, the 'NF-κB signaling pathway' term was significant for both the common and M0:M1*Cn*-exclusive pools of DEGs. This was consistent with the known function of the NF-κB pathway as a regulator of M1 polarization and prior studies by our and other groups showing that *Cn* infection modulates the activity of NF-κB transcription factors in host macrophages [46, 57]. Interestingly, B cell leukemia/lymphoma 2 related protein A1a (*Bcl2a1a*), the murine orthologue of the anti-apoptotic protein A1, was amongst the NF-κB-regulated DEGs present within the M0:M1*Cn*-exclusive pools of DEGs and was upregulated 4.95 fold ($q$ = 0.006); *Bcl2a1a* was also upregulated in *Cn*-infected M2-polarized cells (M0:M2*Cn*, FC = 4.93, $q$ = 0.006). To visualize changes to M1-associated gene interaction networks M0:M1mk and M0:M1*Cn* DEGs were analyzed using STRING (Fig 3B and 3C). Consistent with the notion of disrupted M1 polarization, the gene cluster associated with innate immune function was smaller for the M0:M1*Cn* compared to M0:M1mk DEGs (24 vs. 30 genes), as was the antigen processing and presentation cluster.

Equivalent analyses were performed for the M2 data sets, yielding similar results. Here, of the 583 DEGs associated with M0 to M2 polarization, 234 (~40%) were common to the 340 DEGs from the M0:M2*Cn* comparison (Fig 4A). As before, this indicated that the expression of the remaining 349 (~60%) DEGs associated with the M2 state became more M0-like post-*Cn* infection and the 106 DEGs unique to the M0:M2*Cn* comparison were assumed to be *Cn*-induced changes in gene expression, not associated with either the normal M0 or M2 transcriptional profiles. Similar to the data for M1-polarized cells, genes associated with the KEGG pathway term, 'Phagosome' and 'Antigen processing and presentation' were enriched amongst the common and M0:M2mk unique pools of DEGs. In contrast, genes associated with 'p53 signaling' were only significantly enriched amongst the M0:M2mk pool in the M2 analysis and not in the equivalent M0:M1mk pool. These genes included the canonical p53 targets, *Ccng1*, *Cdkn1a* (p21), and *Mdm2*, all of which were upregulated >3-fold. The pool of M0:M2*Cn*-exclusive DEGs was relatively small and the 'NF-κB signaling pathway' GO term was one of the few that could be associated with this group. This short list included *Plau*, *Cd40*, *Traf1*, and, as previously mentioned, *Bcl2a1a*.

STRING analysis using DEGs from M0:M2mk and M0:M2*Cn* pairwise comparisons also showed differences in the gene cluster associated with innate immune function (Fig 4B). However, in contrast to the analysis of M1-polarized cells, the cluster was larger for M0:M2*Cn* compared to M0:M2mk DEGs (23 vs. 17 genes). Additionally, the antigen processing and presentation cluster was more similar in size for M0:M2mk and M0:M2*Cn* than in the equivalent analysis for M1-polarized cells, suggesting that *Cn* infection possibly has smaller effects on this process in M2 macrophages.

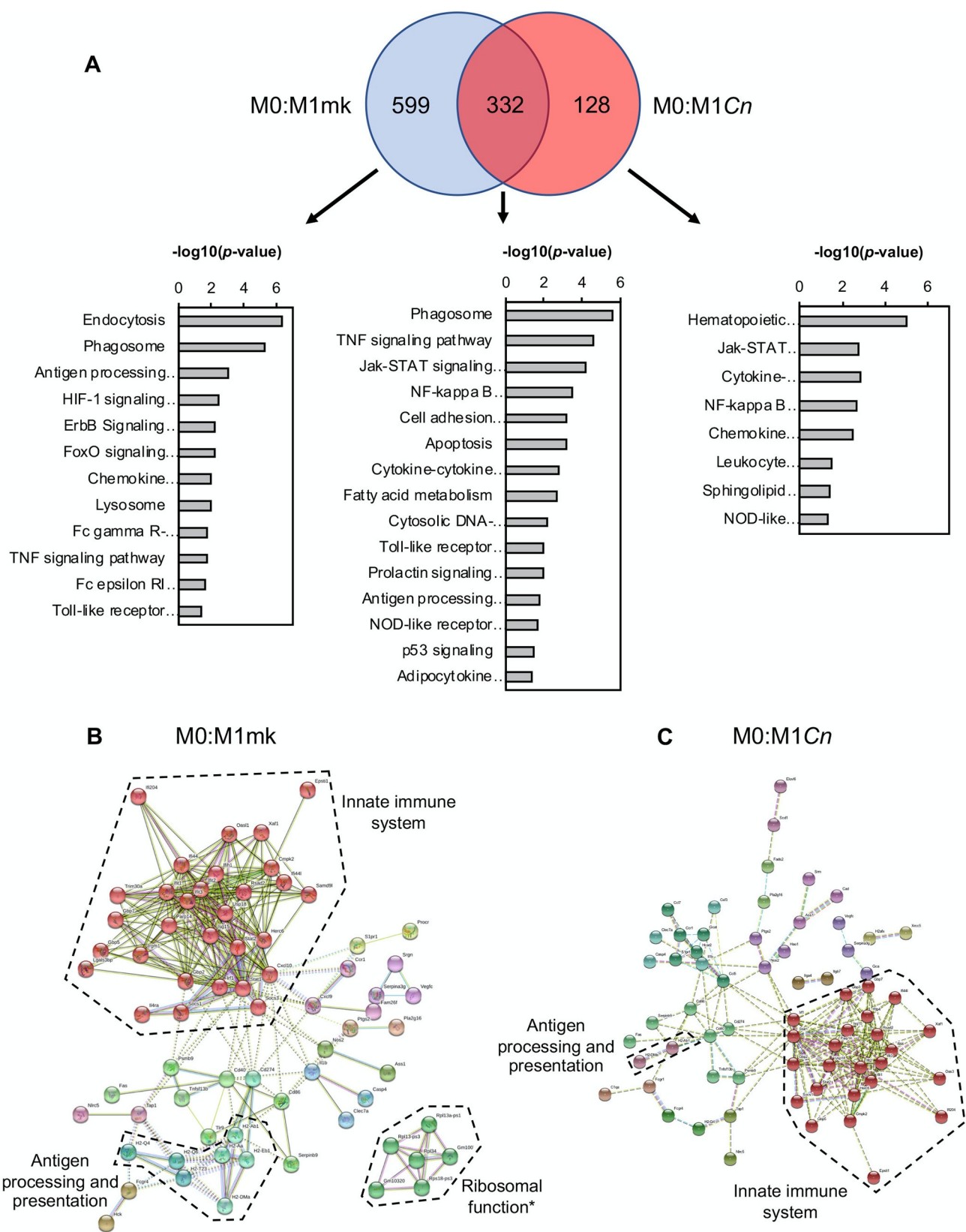

**Fig 3. Gene ontology (GO) analysis of *Cn*-infected M1 macrophages.** (**A**) Venn diagram to represent common and differentially expressed genes in M0:M1mk and M0:M1*Cn* comparisons. These are accompanied by GO analysis performed in DAVID on genes appearing in each division of the Venn diagrams. Relevant pathways are ranked by–log(*p*-value). Differentially expressed genes from (**B**) M0:M1mk and (**C**) M0:M1*Cn* pairwise comparisons were analyzed in STRING. Boundaries enclosing gene clusters with common function are drawn based on gene GO term data and information from the literature. *Note: Many of the genes contained within the 'Ribosome function' boundary are pseudogenes.

## A transcriptome signature of *Cn* infection

Having examined the effects of *Cn* infection on the M1 and M2 transcriptional profile, we sought to determine whether there was a common set of genes affected by *Cn* infection, regardless of host cell polarization state. This was performed by first identifying DEGs from M1mk:M1*Cn* and M2mk:M2*Cn* pairwise comparisons, of which there were 204 and 254, respectively. These data were used to (i) produce a visual representation of gene networks in *Cn*-infected M1 and M2 cells using STRING analysis (S1 Fig), and (ii) identify concordant genes present in both M1mk:M1*Cn* and M2mk:M2*Cn* DEG lists (Tables 3 and 4, respectively). Similar network structures were evident in both M1mk:M1*Cn* (S1A Fig) and M2mk:M2*Cn* (S1B Fig) with both containing clusters associated with the innate immune system and ribosome function, although there were differences in the identity and numbers of genes in these clusters. The importance of the ribosomal function cluster was also questionable as many of these were pseudogenes. The M1mk:M1*Cn* contained a cluster of three genes associated with cell cycle regulation, *Klf4*, *Cdkn1a (p21)*, and *Ccng1*, all of which were downregulated in M1-polarized *Cn*-infected cells. Although *Ccng1* was also downregulated in *Cn*-infected M2 cells, the corresponding cluster was absent, as the expression of *Klf4* and *Cdkn1a* was seemingly unaffected. A cell adhesion cluster could not be identified amongst the M2mk:M2*Cn* DEGs for similar reasons.

In addition to the clustering, these data were also used for KEGG pathway analysis. For the M1mk:M1*Cn* DEGs, an enrichment of genes associated with 'Chemokine signaling pathway' ($p = 0.006$), 'Leukocyte transendothelial migration' ($p = 0.018$), 'Regulation of actin cytoskeleton' ($p = 0.035$), and 'Hematopoietic cell lineage' ($p = 0.036$) was observed. The corresponding list of enriched terms for M2-polarized cells was largely different, containing 'Lysosome' ($p = 0.034$), with only one term, 'Chemokine signaling pathway' ($p = 0.034$), also appearing in the corresponding M2mk:M2*Cn* analysis. However, the list of genes associated with this term for M1 and M2 cells (7 and 6 genes, respectively) also differed with only *Ccl22*, and *Grk5* common between the two polarization states.

Similar analysis using the biological process (BP) GO terms showed an enrichment of genes associated with 'Chemotaxis', 'Endocytosis', and 'Inflammatory response' in *Cn*-infected cells in both polarization states (S1C and S1D Fig). Amongst the various differences between the two polarization states, 'Leukocyte cell-cell adhesion' and 'Toll-like receptor 9 signaling pathway' were only seen in the analysis of M1mk:M1*Cn* DEGs. Conversely, 'Phagocytosis' and 'Positive regulation of phagocytosis' were only seen in the M2mk:M2*Cn* comparison. Interestingly, all genes associated with these two terms (counts of 4 and 5, respectively), were downregulated and included *Pros1*, which encodes Protein S, a regulator of phagocytosis in macrophages [58].

Amongst the DEGs from the M1mk:M1*Cn* and M2mk:M2*Cn* pairwise comparisons, it was found that 38 genes were common and concordant (~15–19%), with 8 upregulated and 30 downregulated (Tables 3 and 4). Interestingly, a large number of these represented reversals or partial reversals of gene expression changes occurring when repolarizing from M0 to M1, M0 to M2, or both. However, seven genes from this set were not part of the M1 or M2 transcriptome profile and appeared unique to *Cn*-infected cells in this analysis. These included three

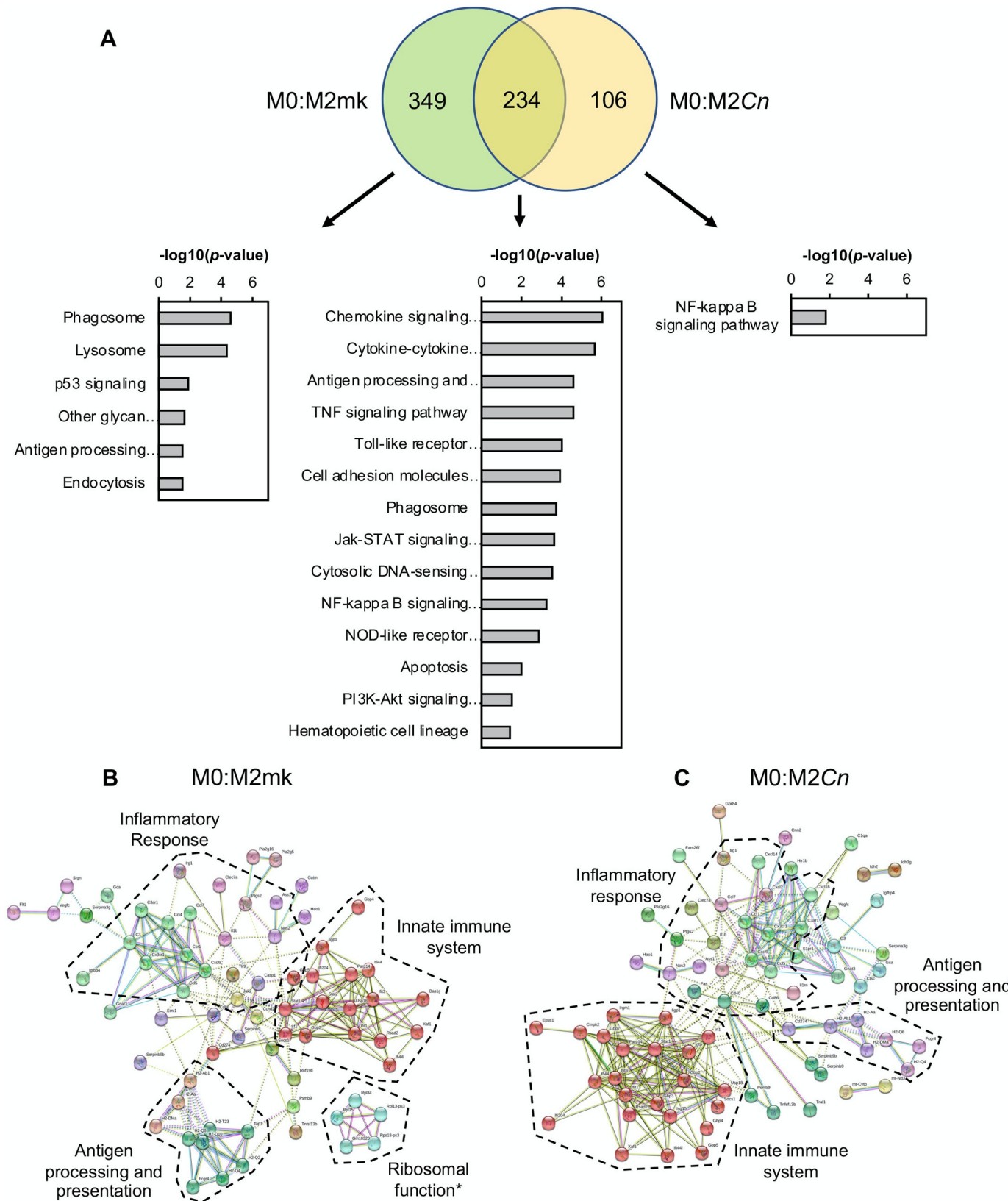

**Fig 4. Gene ontology (GO) analysis of *Cn*-infected M2 macrophages.** (**A**) Venn diagram to represent common and differentially expressed genes in M0:M2mk and M0:M2*Cn* comparisons. These are accompanied by GO analysis performed in DAVID on genes appearing in each division of the Venn diagrams. Relevant pathways are ranked by –log(*p*-value). Differentially expressed genes from (**B**) M0:M2mk and (**C**) M0:M2*Cn* pairwise comparisons were analyzed in STRING. Boundaries enclosing gene clusters with common function are drawn based on gene GO term data and information from the literature. *Note: Many of the genes contained within the 'Ribosome function' boundary are pseudogenes.

upregulated (*Cited1*, *Ccl22*, and *Bcl2a1a*) and four downregulated (*Itgax*, *Ank*, *Lrp1* and *Atp2a2*) genes.

## CITED1 is upregulated by *Cn*-infected M1 and M2 macrophages

The CBP/p300-interacting transactivator with glutamic acid/aspartic acid-rich carboxyl-terminal domain (*CITED*) gene family encodes transcriptional co-regulators that activate or repress gene expression through direct interaction with CBP/p300. Of the three *CITED* family members present in mammals (1, 2, and 4), only *CITED2* appears to be abundantly expressed in human and murine macrophages [59].

While the expression of *CITED2* transcripts were unchanged in the conditions tested in this study, *CITED1* showed the largest fold-change of all concordant DEGs and was upregulated in both M1 and M2 *Cn*-infected cells (FC = 14.81 and 20.49 in M1 and M2 cells, respectively). To determine whether this change in transcript abundance was accompanied by a similar increase at the protein level, western blotting was used to measure CITED1 levels in M1 and M2-polarized *Cn*-infected cells. CITED1 protein levels were strongly increased in post-*Cn* infection in M1 polarized but not M2 polarized macrophages or mock-infected controls (Fig 5A and 5B). Additionally, under conditions where macrophages exhibited much-reduced rates of phagocytosis (i.e. naive macrophages or cells treated with IL-4 alone without prior IFNγ stimulation), exposure to *Cn* did not stimulate increased CITED1 expression. Taken together, these data indicated that CITED1 was expressed in response to intracellular rather than extracellular *Cn* and was not affected by polarization alone. As all previous experiments were performed using a relatively high concentration of *Cn* (MOI 3:1) with cells harvested at 24 h post-infection, a final experiment was performed to determine whether Cited1 was induced at lower *Cn* concentrations and earlier timepoints. Macrophages were infected with opsonized *Cn* at a concentration of 1, 2, or 3 *Cn* per macrophage and harvested at 6 h post-infection. Here, there was a dose-dependent response for iNos (Fig 5C and 5D), similar to that observed in previous studies [17]. Similarly, higher yeast concentrations stimulated increased Cited1 protein expression (Fig 5C and 5E).

**Table 3. Common concordant upregulated genes in *Cn*-infected cells.**

| Gene | M1mk vs. M1*Cn* | | M2mk vs. M2*Cn* | |
|---|---|---|---|---|
| | FC | *q*-value | FC | *q*-value |
| Cited1 | 14.81 | 6.26E-03 | 20.49 | 1.61E-02 |
| Hsf3 | 7.93 | 2.86E-02 | 8.04 | 0.0200165 |
| Jarid2 | 5.45 | 2.52E-02 | 2.88 | 0.0389813 |
| Tmtc2 | 5.14 | 9.61E-03 | 3.94 | 0.00626191 |
| Ccl22 | 5.09 | 3.50E-02 | 4.1 | 0.00626191 |
| Sspn | 3.66 | 4.28E-02 | 3.21 | 0.0251778 |
| Wdr89 | 3.09 | 3.34E-02 | 2.43 | 0.0389813 |
| Bcl2a1a | 2.68 | 4.68E-02 | 2.95 | 0.0171929 |

Top common concordant upregulated genes in M1 and M2 *Cn*-infected macrophages (FC = Fold-change)

**Table 4. Common concordant downregulated genes in *Cn*-infected cells.**

| Gene | M1mk vs. M1*Cn* | | M2mk vs. M2*Cn* | |
|---|---|---|---|---|
| | FC | *q*-value | FC | *q*-value |
| Sorl1 | 37.12 | 3.90E-02 | 3.86 | 0.046023 |
| Dusp6 | 14.54 | 1.29E-02 | 6.72 | 0.0375289 |
| C5ar1 | 12.07 | 6.26E-03 | 5.45 | 0.00626191 |
| Ehd2 | 11.63 | 6.26E-03 | 5.08 | 0.00626191 |
| Grk5 | 8.83 | 6.26E-03 | 4.48 | 0.0128586 |
| Cd300ld | 7.24 | 6.26E-03 | 2.7 | 0.0389813 |
| Zfp146 | 6.31 | 1.61E-02 | 3.97 | 0.0304857 |
| Endod1 | 6.21 | 6.26E-03 | 2.57 | 0.027269 |
| Usp9x | 6.11 | 2.00E-02 | 3.88 | 0.0383977 |
| Xdh | 6.02 | 6.26E-03 | 3.21 | 0.0171929 |
| Stom | 4.86 | 6.26E-03 | 2.84 | 0.00626191 |
| Pik3ap1 | 4.48 | 6.26E-03 | 3.52 | 0.0389813 |
| Car5b | 4.36 | 2.52E-02 | 3.91 | 0.034954 |
| Itgax | 4.2 | 2.25E-02 | 3.73 | 0.0171929 |
| Gatm | 4.06 | 6.26E-03 | 7.79 | 0.00626191 |
| Ms4a6b | 3.46 | 3.50E-02 | 4.81 | 0.0128586 |
| Fgd3 | 3.42 | 1.29E-02 | 2.64 | 0.0383977 |
| Man2b1 | 3.27 | 1.29E-02 | 3.37 | 0.00961334 |
| Lamc1 | 3.23 | 3.63E-02 | 3.69 | 0.0200165 |
| Tep1 | 3.1 | 4.60E-02 | 2.74 | 0.0375289 |
| Dhx40 | 3.07 | 4.77E-02 | 3.22 | 0.0322409 |
| Ank | 3.07 | 2.00E-02 | 2.97 | 0.00626191 |
| Ccng1 | 2.9 | 2.00E-02 | 2.89 | 0.00626191 |
| Stt3a | 2.9 | 2.73E-02 | 2.81 | 0.0249383 |
| Plekho2 | 2.88 | 3.84E-02 | 3.36 | 0.0251778 |
| Cdc42se2 | 2.82 | 2.86E-02 | 2.39 | 0.0383977 |
| Lrp1 | 2.75 | 4.77E-02 | 2.55 | 0.0357649 |
| Myo1e | 2.7 | 3.63E-02 | 3.75 | 0.00626191 |
| Nckap1l | 2.63 | 3.75E-02 | 2.52 | 0.0251778 |
| Atp2a2 | 2.5 | 4.28E-02 | 2.23 | 0.0420558 |

Top common concordant downregulated genes in M1 and M2 *Cn*-infected macrophages (FC = Fold-change)

## Discussion

The polarization of macrophages is highly plastic, shifting along a continuum of functionally distinct states during different stages of an infection [60]. While these changes are largely driven by the milieu of cytokines and inflammatory regulators present within the microenvironment of the cell, a broad range of pathogens are known to subvert this process as part of intracellular survival strategies, often by modulating the activity of STAT1, 3, and 6, the primary controllers of macrophage polarization [61]. While the primary driver of macrophage polarization during *Cn* infections would appear to be the changing balance of Th1 and Th2 cytokines [62], there is evidence to suggest that direct interaction with *Cn* influences the macrophage polarization state. In two separate *in vitro* studies, *Cn* was shown to suppress NO expression in macrophage cell lines, most likely through inhibition of *Nos2* expression, resulting in an M2-like state [63, 64]. In a more recent study using RAW264.7 cells, co-culture with opsonized *Cn* promoted a weak M1 phenotype and increased expression of *Nos2*, which was

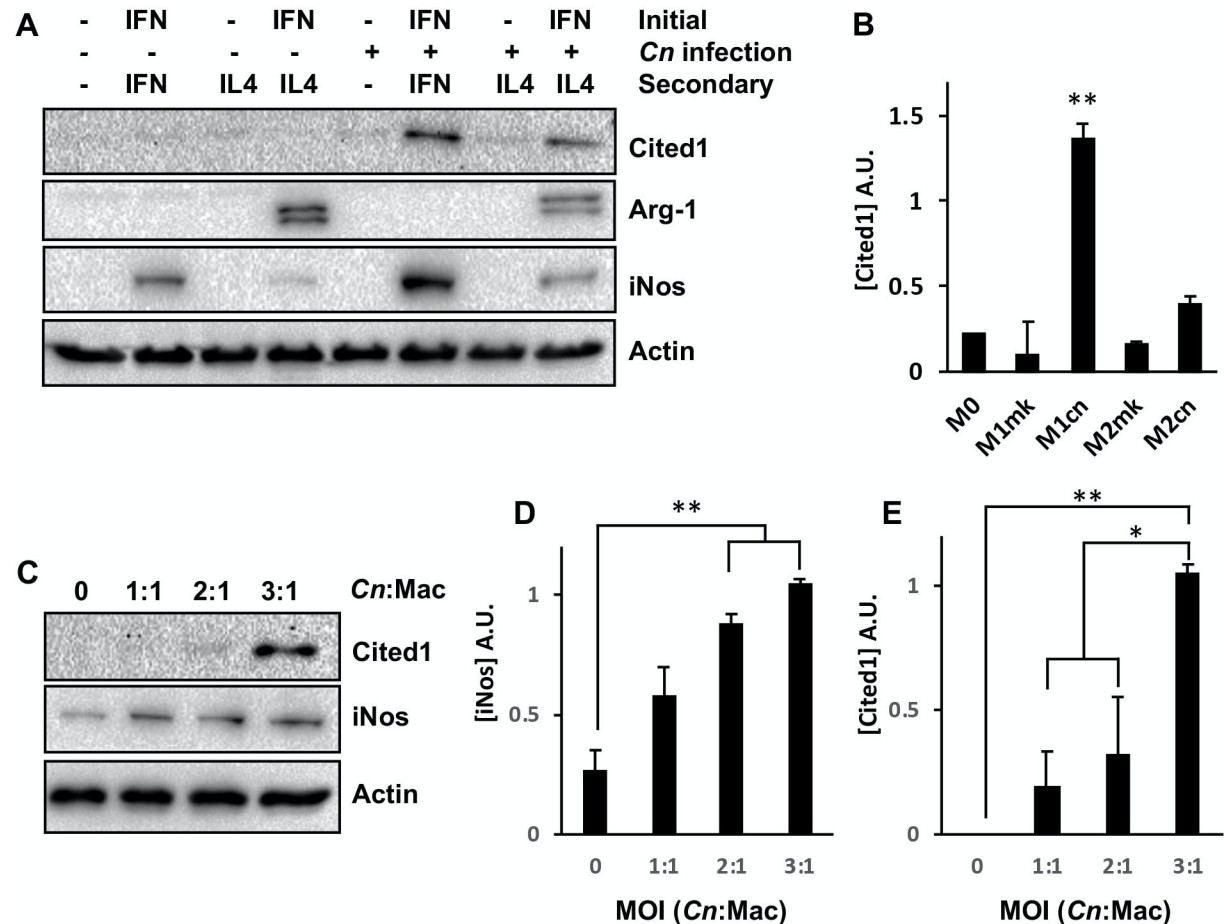

**Fig 5. *Cn* infection stimulates increased Cited1 expression in M1 and M2-polarized macrophages. (A)** Western blot analysis of RAW264.7 macrophages for Cited1 and the M1 (iNos) and M2 (Arg-1) markers. Cells were treated as indicated and harvested at 24 h post-infection. **(B)** Quantification of Cited levels in *(A)*. **(C)** Western blot analysis of RAW264.7 macrophages for Cited1 and iNos at 6 h post-infection at the indicated MOI. Quantification of **(D)** iNos and **(E)** Cited1 levels in *(C)*. For *(B, D, +E)* densitometry is based on three discrete biological repeats. Error is represented as S.E. Statistical differences between samples were appraised by one-way ANOVA followed by a Tukey's multiple comparison test. Statistical significance is indicated as follows: *, $p < 0.05$; **, $p < 0.01$.

reversible on exposure to IL-4 [17]. However, macrophage polarization was interrogated in each of these studies by measuring the expression of a small number of transcripts or markers of M1 and M2 polarization rather than studying the broader gene expression networks associated with each state.

This study is the first to comprehensively examine the effect of intracellular *Cn* infection on the transcriptome of M1- and M2-polarized cell line macrophages using an RNA sequencing-based approach. In agreement with earlier studies [17], we also find that *Cn* infection has relatively modest effects on the principle polarization markers, including *Nos2* and *Arg1*. A small reduction in *Arg1* transcript levels in *Cn*-infected M2 cells was observed that did not extend to the protein level (Fig 2B and 2C). Conversely, although *Nos2* transcript levels were not affected by *Cn*-infected M1 macrophages, there was a 40% increase in protein levels (Fig 2B–2D). These discrepancies suggest additional regulatory influences, such as translational or post-translational mechanisms regulating the translation and/or stability of the two proteins.

However, these data revealed an extensive disruption of both the M1 and M2 transcriptome in *Cn*-infected macrophages, shifting cells towards or actively retaining them at a more

M0-like state. Given that culture conditions were tightly controlled throughout these experiments with growth medium and cytokines replaced at regular intervals, we believe that these changes are genuine, are actively employed, and not simply caused by relaxation back to an M0 state, as might happen on withdrawal of polarizing stimuli. Additionally, a common set of genes that were affected in a similar fashion regardless of host cell polarization state were identified and constitute a transcriptional signature of intracellular *Cn* infection (Tables 3 and 4).

The effect of *Cn* exposure on the transcriptome of monocytes and macrophages has been investigated in a number of earlier studies [24, 26, 65–67]. While the overlap between the gene expression changes detected in these studies and the current study is limited, possibly due to the use of differing cell models and in some cases, older microarray technology [26], certain consistencies are evident. When regarded from the level of signaling pathways affected rather than individual genes, both this study and the Coehlo *et al* study detect changes in the activity of the HIF-1 signaling pathway (Fig 3A), which has also been observed in pulmonary fungal infections [68]. Additionally, NF-κB, JAK-STAT, TNF, and Toll-like receptor (TLR) signaling pathway are also amongst the highest ranked KEGG GO terms in both this study and the Chen *et al* study [24] (Figs 3 and 4). This was unsurprising as *Cn* infection has been previously shown to affect each of these pathways. In brief, intracellular *Cn* has been shown to affect both canonical and noncanonical NF-κB signaling in host cells to stimulate fungal-induced apoptosis and arrest the cell cycle [46, 57]. STAT1 has been shown to be important for successful clearance of *Cn* and M1 macrophage polarization in a murine model with macrophage-specific STAT1 knockdown and H99γ, a *Cn* strain engineered to express IFNγ [69, 70]. Additionally, TLR signaling has been shown to play an important role in macrophage:*Cn* interactions in numerous studies. Both TLR2 and 4 are known to interact with the major *Cn* capsular polysaccharide, glucuronoxylomannan [71, 72], and TLR2 and the downstream adaptor protein, MyD88, have been shown to have important roles in the anti-fungal response to *Cn* infection in mice [73–75]. As a final note on the similarity between the affected pathways in this study and the Chen *et al* study; their experiments used heat-killed rather than live *Cn*. This implies that many of the changes observed may not require *Cn* metabolic activity. One possible explanation for this might be the immunomodulatory effects of capsular polysaccharides, which are present on the surface of live and dead ingested yeast [46, 76]. This is likely true for GXM-stimulated TLR and NF-κB signalling [46, 76].

These data and analyses also allowed the identification of common and differing cellular processes affected by *Cn* infection in M1 and M2 cells. The most notable common effect was the reversal of transcriptome changes associated with phagocytosis and lysosomal function (Figs 3A and 4A). For example, transcripts encoding lysosomal components, including LAMP2 and various lysosomal hydrolases that were upregulated upon polarization to M1 or M2, were returned to M0-levels in *Cn*-infected cells. We also Decreased expression of various subunits of the V-type proton pump required for lysosome acidification (e.g. *Atp6v0a2* and *Atp6v1d*) was also observed. However, *Atp6v0d2*, encoding a macrophage-specific component of the pump [77], was amongst the top upregulated genes in *Cn*-infected M1-polarized cells (M1mk:M1*Cn* FC = 12.98, *q* = 0.036). This particular subunit has been shown to have a role in enhancing autophagosome-lysosome fusion as part of the response to *S*. Typhimurium infection [77]. Therefore, it seems plausible that *Cn* may promote an exchange of lysosomal proton-pump components as part of the response to intracellular infection.

An additional commonality in the responses of M1 and M2 cells to *Cn* infection was the altered activity of the NF-κB signaling pathway (Figs 3A and 4A). This was a common GO term enriched in the pool of DEGs associated with M1 and M2 *Cn*-infected cells. Indeed, it was the only one identified for M2 cells (Fig 4A). However, with the exception of the NF-κB-regulated gene, *Bcl2a1a*, which has not previously been linked to intracellular infection by *Cn*,

there was no overlap between the genes associated with this term in the two states, suggestive of unique effects on the pathway in differing polarization states.

Another notable difference between the polarization states was the presence of 'p53 signaling' as an enriched GO term for M2mk but not M1mk-exclusive genes (Figs 3A and 4A). To our knowledge, changes in p53 signaling have not previous been associated with *Cn* infection of macrophages. This result is potentially meaningful as p53 suppresses the M2 phenotype *in vivo* by downregulating the expression of M2-associated genes [78]. While this might suggest that this inhibition was removed or weakened post-*Cn* infection, the relatively small difference in M2 marker expression (e.g. *Arg1*) and the re-establishment of a more M0-like state indicated this may not be the case.

Perhaps the most significant outcome of this study was the identification of a transcriptome signature of *Cn*-infection, a set of genes commonly and concordantly regulated in both polarization states. This included the downregulation of 30 genes and upregulation of a relatively small pool of 8 functionally diverse genes (Tables 3 and 4). This latter group included the transcriptional regulators *Cited1*, *Hsf3*, and *Jarid2*, the cytokine *Ccl2*, and the anti-apoptotic factor *Bcl2a1a*. Of these, *Cited1*, which was previously known as melanocyte-specific protein 1 (*Msg1*), showed the largest fold change and was also upregulated at the protein level post-*Cn* infection in M1-polarized macrophages (Fig 5).

As a transcriptional co-regulator, CITED1 proteins cannot bind DNA directly and interact with gene enhancer elements by protein-protein interactions with other transcription factors, recruiting CBP/p300 to regulate gene expression [79]. To date, Cited1 has been shown to co-regulate estrogen receptor alpha [80], TGF-β$^4$/Smad4 [81], and Wnt/β-catenin-responsive genes [80–83]. While the specific function of CITED1 in macrophages remains enigmatic and has yet to be thoroughly investigated, CITED2 has been shown to repress proinflammatory gene expression associated with M1-polarizing stimuli, is itself induced by IL-4 and IL-13, and enhances the expression of M2-associated genes [59]. This is likely achieved through destabilization of HIF1α proteins [78] and the attenuation of canonical NF-κB transcription factor activity [84]. This raises the interesting possibility that Cited1 may serve a similar function during fungal infection. This is the subject of ongoing investigations in the lab and the findings will be described in detail as part of a future publication.

In addition to *Cited1*, another notable common and concordant *Cn*-responsive gene identified in the analysis was *Bcl2a1a*, the murine orthologue of the human anti-apoptotic Bcl-2 family member A1. *Bcl2a1a* is known to protect immune cells from TNFα-induced apoptosis, and its expression is directly regulated by NF-κB [85]. Given the damage wrought by intracellular *Cn* growth on host macrophages and the surprisingly low rates of cell death and apoptosis observed [26], it is perhaps unsurprising that the list of common and concordant *Cn*-responsive genes included anti-apoptotic genes. However, its significance in the context of this study remains unclear, particularly as the 'apoptosis' GO term appeared only in the pools of DEGs common to mock- and *Cn*-infected cells of both polarization states.

In addition to identifying previously unrecognized *Cn*-stimulated gene expression changes in host macrophages, this study also helps to demonstrate the importance of carefully controlling nutrient levels in *in vitro* culture systems used to study the *Cn*:macrophage interaction, especially if live rather than heat-killed *Cn* are utilized and experiments last more than a few hours. In preliminary experiments that preceded this study, a significant reduction in the expression of the glucose-responsive gene *TXNIP* was detected at the transcript and protein level (Fig 1G). As TXNIP is a regulator of p53, NF-κB, and other pathways associated with the control of macrophage polarization, glucose depletion may impact macrophage phenotype and result in spurious gene expression changes that do not reflect the potential effects of intracellular *Cn* growth on the transcriptome of host macrophages in an *in vivo* setting [86–88].

Regular replacement of culture medium in these experiments prevented loss of *TXNIP* expression in the RNA sequencing experiments described in this publication, and we strongly recommend that future *in vitro* studies account for this during experimental design.

As a final note of caution, the work described here was performed using RAW 264.7 cells, a murine-leukaemia macrophage-like cell line. While these cells have been shown to mimic the responses of bone marrow-derived macrophages to challenge with microbial ligands [89], and exhibit similar polarization state plasticity [17], they are phenotypically distinct from primary macrophages and may not necessarily respond in an identical fashion during *Cn* infection. Macrophage cell lines can exhibit a more restricted and slower transcriptional response to intracellular pathogens [90], and they might best be considered as a useful but rudimentary macrophage model for the study of innate immune responses to microbial challenge. For this reason, future studies will be conducted using primary macrophages to build upon this initial foundational analysis and gain a clearer picture of the *in vivo* macrophage response to *Cn*.

## Supporting information

**S1 Fig. STRING analysis of gene networks associated with *Cn* infection of M1- and M2-polarized macrophages DEGs from (S1A) M1mk:M1*Cn* and (S1B) M2mk:M2*Cn* pairwise comparisons were analyzed in STRING.** Boundaries enclosing gene clusters with common function were drawn based on gene GO term data and information from the literature. *Note: Many of the genes contained within the 'Ribosome function' boundary are pseudogenes. GO analysis performed in DAVID on DEGs from (**S1C**) M1mk:M1*Cn* and (**S1D**) M2mk:M2*Cn* pairwise comparisons. Relevant pathways are ranked by–log($p$-value).
(TIFF)

**S1 File. RNA sequencing data–all conditions compared to M0 pairwise comparisons of M0 to all other conditions.**
(XLSX)

**S2 File. RNA sequencing data–mock vs. infected pairwise comparisons of mock- and *Cn*-infected samples.**
(XLSX)

**S3 File. Uncropped western blots original, unadjusted, and uncropped western blot images.**
(PDF)

## Acknowledgments

The authors would like to thank Kirsten Cunningham and Dan Bryant for useful discussions and assistance with experiments associated with this study.

## Author Contributions

**Conceptualization:** Erin E. McClelland, Rebecca L. Seipelt-Thiemann, David E. Nelson.

**Data curation:** Rebecca L. Seipelt-Thiemann.

**Formal analysis:** Niah Frantzen, James Mendez, Rebecca L. Seipelt-Thiemann, David E. Nelson.

**Funding acquisition:** Erin E. McClelland, Rebecca L. Seipelt-Thiemann, David E. Nelson.

**Investigation:** Aarthi Subramani, Prianca Griggs, Niah Frantzen, Jamila Tucker, Jada Murriel, Linda M. Sircy, Grace E. Millican, Erin E. McClelland, Rebecca L. Seipelt-Thiemann, David E. Nelson.

**Methodology:** Aarthi Subramani, Erin E. McClelland, Rebecca L. Seipelt-Thiemann.

**Project administration:** David E. Nelson.

**Resources:** Erin E. McClelland.

**Supervision:** Erin E. McClelland, Rebecca L. Seipelt-Thiemann, David E. Nelson.

**Visualization:** Niah Frantzen, James Mendez, Rebecca L. Seipelt-Thiemann.

**Writing – original draft:** Aarthi Subramani, Prianca Griggs, Niah Frantzen, Erin E. McClelland, Rebecca L. Seipelt-Thiemann, David E. Nelson.

**Writing – review & editing:** Aarthi Subramani, Prianca Griggs, Niah Frantzen, James Mendez, Jamila Tucker, Jada Murriel, Linda M. Sircy, Erin E. McClelland, Rebecca L. Seipelt-Thiemann, David E. Nelson.

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
