## [Decision Letter · Decision Letter 0]

18 Jun 2020

PONE-D-20-13968

Intracellular *Cryptococcus neoformans* disrupts the transcriptome profile of M1- and M2-polarized host macrophages

PLOS ONE

Dear Dr. Nelson,

Thank you for submitting your manuscript to PLOS ONE. After careful consideration, we feel that it has merit but does not fully meet PLOS ONE’s publication criteria as it currently stands. Therefore, we invite you to submit a revised version of the manuscript that addresses the points raised during the review process.

We look forward to receiving your revised manuscript.

Kind regards,

Michal A Olszewski, DVM, PhD

Academic Editor

PLOS ONE

Journal Requirements:

2. We note that you are reporting an analysis of a microarray, next-generation sequencing, or deep sequencing data set. PLOS requires that authors comply with field-specific standards for preparation, recording, and deposition of data in repositories appropriate to their field. Please upload these data to a stable, public repository (such as ArrayExpress, Gene Expression Omnibus (GEO), DNA Data Bank of Japan (DDBJ), NCBI GenBank, NCBI Sequence Read Archive, or EMBL Nucleotide Sequence Database (ENA)). In your revised cover letter, please provide the relevant accession numbers that may be used to access these data. For a full list of recommended repositories, see http://journals.plos.org/plosone/s/data-availability#loc-omics or http://journals.plos.org/plosone/s/data-availability#loc-sequencing.

We note that one or more of the authors are employed by a commercial company: M&P Associates, Inc..

3.1. Please provide an amended Funding Statement declaring this commercial affiliation, as well as a statement regarding the Role of Funders in your study. If the funding organization did not play a role in the study design, data collection and analysis, decision to publish, or preparation of the manuscript and only provided financial support in the form of authors' salaries and/or research materials, please review your statements relating to the author contributions, and ensure you have specifically and accurately indicated the role(s) that these authors had in your study. You can update author roles in the Author Contributions section of the online submission form.

3.2. Please also provide an updated Competing Interests Statement declaring this commercial affiliation along with any other relevant declarations relating to employment, consultancy, patents, products in development, or marketed products, etc. 

Reviewers' comments:

Reviewer's Responses to Questions

**Comments to the Author**

1. Is the manuscript technically sound, and do the data support the conclusions?

Reviewer #1: Yes

Reviewer #2: Partly

2. Has the statistical analysis been performed appropriately and rigorously? 

Reviewer #1: Yes

Reviewer #2: I Don't Know

3. Have the authors made all data underlying the findings in their manuscript fully available?

Reviewer #1: Yes

Reviewer #2: Yes

4. Is the manuscript presented in an intelligible fashion and written in standard English?

Reviewer #1: Yes

Reviewer #2: Yes

5. Review Comments to the Author

Reviewer #1: In the manuscript Subramani et. al. investigated the effects of intracellular C. neoformans on the transcriptome profile of M1 and M2 macrophages. In vitro experiment showed clearly that while polarizing cytokines could induce or repolarize M1/M2, Cn infection cases relatively small changes in the gene expression of classical M1/M2 markers such as Nos2 or Arg1. They also identified a gene cluster induced by cn infection that is common to both polarization statuses. The experiments are well designed and the manuscripts are well written. It is a very nice piece of work. The reviewer only has several minor concerns for the authors to address.

1. There are wrong labels in Fig. 1C and 1D

2. The authors identified 204 and 254 DEGs from the M1mk:M1Cn and M2mk:M2Cn pairwise comparisons, respectively. However, they mainly focused on the 38 genes that were common and concordant regulated. I think information about the other DEGs that Cn regulates in M1 or M2 conditions are also interesting and showed be reported and summarized in the results.

Reviewer #2: In this manuscript by Subramani et al, the authors describe transcriptional changes induced in cell line macrophages after exposure to Cryptococcus neoformans. These data are interesting, but there are some issues with the included experiments as well as some additional studies that would make the study more complete.

Major points:

1. The studies in this manuscript are interesting, especially with the discovery of transcriptional regulators such as CITED1. However, using cell lines does not always give an accurate picture of what happens in primary macrophages (especially lung macrophages). Therefore, the authors should verify these data using lung macrophages ex vivo and/or using in vivo mouse pulmonary cryptococcal studies.

2. In the Introduction, page 4, lines 85-86, the authors mention that their studies found altered activity in JAK-STAT and NF-kB pathways. This is already known (published in several papers on crypto and macrophage activation), so why is this a novel finding?

3. In the methods section, page 5, line 103, the authors use a high concentration of C. neoformans in these studies. It would be helpful to have multiple concentrations/MOI of C. neoformans in the assay, since different ratios of crypto to macrophages can make a big difference in outcome.

4. The studies discussed in the results on page 13, lines 255-265 seem unnecessary, as we already know that cell line macrophages can be polarized using in vitro stimuli. Therefore, this section (and the accompanying data) should be removed from the manuscript.

5. In the discussion, these data should really be put more into context with the current Cryptococcus literature – not just the general macrophage literature or bacterial pathogen literature.

Minor points:

1. In the introduction, page 3 lines 59-60, there are several more C. neoformans papers on macrophage activation and C. neoformans that have been published over the last 10+ years. The authors should reference a wider range of papers. To find these, in pubmed, search “macrophage activation Cryptococcus” or “macrophage M1 M2 Cryptococcus.”

2. In the introduction, page 4, line 77 – why not reference a Cryptococcus paper? There are many papers and reviews on C. neoformans specifically available. Fungi are different than bacteria, so a fungal-specific paper would be more appropriate than a bacterial paper here.

3. In the introduction, page 5, line 87, the authors mention a M0-like state, but this is not defined anywhere in the manuscript. Please clarify what is meant by M0.

4. In the methods section, page 6, line 118, the authors state that cells were given IFN-g to “maintain the M1 polarization state.” How was this maintenance verified? This information needs to be added to the methods section.

5. In the results section, page 14, lines 287-290, the authors mention replacing the medium throughout the experiment at 6h intervals. This seems like quite a bit of manipulation – was this methodology compared to controls without media changes? If not, these studies need to be conducted as a control for the manipulation mentioned in the text.

6. In the results section, page 14, line 293, the authors mention that conditions were optimized to maximize the Cn-infected M1 and M2 polarized macrophages. How was the percent infected determined? Microscopy? Flow? There is no mention of these studies in the methods, so please include these details. Similarly, this is also discussed on page 16, line 297. Again, nothing is mentioned about how the infection efficiencies were calculated. This information needs to be added to the methods section.

7. In the legend for Figure 4 (page 21), please add the number of experiments represented by these data.

8. In the Discussion, pages 26-27, lines 499-503, the authors reference a Salmonella paper in regards to signaling in M1/M2 polarization. It would be helpful to also include what is known about signaling from the Cryptococcus literature – there are several papers that can be referenced.

9. In the Discussion, page 27, line 512, please indicate “cell line macrophages” and not simply “macrophages.” In addition, in the same section (lines 511-520), it would be helpful to also explain the discrepancies caused by using only a cell line instead of primary macrophages in the study.

10. In the Discussion, page 28, line 530, there are definitely more than 2 references for Cryptococcus studies investigating gene expression in macrophages following interaction with C. neoformans. Include the relevant studies here. Similarly, in line 538, there is more than a single study showing the activity of these signaling pathways in macrophages exposed to C. neoformans.

6. PLOS authors have the option to publish the peer review history of their article (what does this mean?). If published, this will include your full peer review and any attached files.

Reviewer #1: No

Reviewer #2: No

---

## [Author Response · Author response to Decision Letter 0]

31 Jul 2020

Responses to issues raised by referees

Reviewer 1:

“There are wrong labels in Fig. 1C and 1D.”

We apologize for these errors. This has now been corrected.

 “The authors identified 204 and 254 DEGs from the M1mk:M1Cn and M2mk:M2Cn pairwise comparisons, respectively. However, they mainly focused on the 38 genes that were common and concordant regulated. I think information about the other DEGs that Cn regulates in M1 or M2 conditions are also interesting and showed be reported and summarized in the results.”

 We thank the reviewer for suggesting this. This has now been added to the results section on pages 23-24, lines 561-580 and is accompanied by a new figure summarizing GO analysis of these DEGs from the M1mk:M1Cn and M2mk:M2Cn pairwise comparisons. We highlight overlapping enriched terms between these two lists (e.g. ‘Chemokine signaling pathways’) and comment on notable differences.

Reviewer 2:

“The studies in this manuscript are interesting, especially with the discovery of transcriptional regulators such as CITED1. However, using cell lines does not always give an accurate picture of what happens in primary macrophages (especially lung macrophages). Therefore, the authors should verify these data using lung macrophages ex vivo and/or using in vivo mouse pulmonary cryptococcal studies.”

 We agree that the behavior of macrophage cell lines may not always match that of ex vivo macrophages and we have every intention of following up this work with subsequent studies in primary cells. However, due to the pandemic, the temporary shutdown of our laboratory facilities, and the ongoing disruption this is causing to our research efforts, we do not believe that we can complete this work in a timely fashion. We assert that the manuscript serves as a complete and publishable unit of work in its current form that would be of interest to the macrophage and Cryptococcus research community even without the primary cell data. As suggested in the reviewer’s minor comments, we have added new text to the discussion to make it clear that the work was performed exclusively in cell lines and describes the limitations associated with this (pages 35-36, lines 872-887). 

“In the Introduction, page 4, lines 85-86, the authors mention that their studies found altered activity in JAK-STAT and NF-kB pathways. This is already known (published in several papers on crypto and macrophage activation), so why is this a novel finding?”

 We agree with the reviewer that this is not novel. The role of JAK-STAT signaling, particularly STAT1, has been explored extensively in a series of articles from Floyd L. Wormley Jr’s lab. Additionally, NF-κB modulation by Cn has previously been investigated by various labs, including our own. In the current manuscript, we did identify NF-κB-regulated genes induced in Cn infected cells, such as the Bcl2a1a, which encodes the murine orthologue of the anti-apoptotic Bcl-2 family protein, A1. Expression of this gene has not previously been linked with Cn infection and is therefore novel. However, we do concede that the line identified by the reviewer, as it appears in the final research summary paragraph of the introduction, may suggest to readers that our finding of altered JAK-STAT and NF-κB was entirely novel even though we did not explicitly say that it was. We have no desire to mislead readers and so we have removed this line. Additionally, we have added new text to the discussion with references concerning the importance of JAK-STAT and NF-κB in the discussion. These changes are described in our response to minor comment 10 from the second reviewer.

“In the methods section, page 5, line 103, the authors use a high concentration of C. neoformans in these studies. It would be helpful to have multiple concentrations/MOI of C. neoformans in the assay, since different ratios of crypto to macrophages can make a big difference in outcome.”

 Our rational for using the 3:1 yeast:macrophage MOI was to insure the highest percentages of macrophages were infected for our RNAseq experiments as we were interested in measuring the effect of intracellular Cn on gene expression. Using a lower Cn concentration would have increased the proportion of the RNA assayed in these experiments that originated from uninfected cells and therefore reduced our ability to detect the transcriptional changes that our experiments were designed to detect. However, we completely agree with the reviewer’s comment and have incorporated new data in the revised manuscript comparing iNos and Cited1 protein expression in cells infected with 1, 2, and 3 Cn per macrophage. This is included as Figure 5C-E in the revised manuscript. As was previously shown in Davis et al, we also see a dose-dependent effect on iNos levels. Similarly, we were able to detect Cited1 expression at lower Cn concentrations, but it was highest at a 3:1 MOI. Notably, this experiment was carried out at 6 rather than 24 h post-infection and therefore required no medium changes. It also shows that Cited1 was expressed far earlier than we previously thought. This is important as it raises the possibility that Cited1 itself, a transcription co-regulator, may have a role in some of the transcriptional changes we observed at 24 h post-transfection in our RNAseq experiments.

“The studies discussed in the results on page 13, lines 255-265 seem unnecessary, as we already know that cell line macrophages can be polarized using in vitro stimuli. Therefore, this section (and the accompanying data) should be removed from the manuscript.”

 As the reviewer suggests, it is already known that cell line macrophages can be repolarized using in vitro stimuli (1). However, we are using these data to make a different and important point about the polarization of RAW264.7 cells specifically. We show a difference in the expression of Arg-1 proteins. The levels of Arg-1 proteins, a principle marker of the M2 state, are substantially higher (>5 fold) if cells are repolarized from the M1 state rather than polarizing directly from the naïve/M0 state. While the approach taken for this experiment was similar to the transcript level (RT-PCR) experiment published as Figure 6A in Davis et al (1), the result is inherently different. In those experiments, no difference was observed in the expression of Arg1 transcripts between cells directly polarized from the naïve state and those previously treated with IFNγ prior to IL-4. While Davis et al do also provide immunofluorescent data looking at Arg1 protein expression under similar conditions, the data is not quantified, and the result appears different from ours. This does not mean to say that we believe the Davis et al result is wrong. We do not. We feel that it is more likely that additional post-transcriptional and post-translational mechanisms are in play that may help to explain the difference in our protein level and their transcript level data.

 We believe this discrepancy is important for several reasons; first, we feel that it is important information for others who intend to use RAW264.7 cells as a macrophage model in experiments requiring M2 polarization, as our data indicates that expression of Arg1 transcripts, as seen after stimulation with IL-4 alone, may not result in a corresponding increase in Arg1 protein. Second, we believe that these data help to justify the experimental approach we took when preparing samples for the RNAseq experiments where we M1-polarized all cells prior to infection before maintaining M1 or repolarizing to M2. For these reasons, we strongly believe that the text and figure should be retained in the manuscript. However, if the reviewer insists that it should be removed we would be happy to move it to the supplementary material.

“In the discussion, these data should really be put more into context with the current Cryptococcus literature – not just the general macrophage literature or bacterial pathogen literature.”

 On reflection, we agree that the original discussion was not sufficiently focused on the Cryptococcus literature. Where possible, we have removed unnecessary text and citations concerning bacterial pathogens (one on Salmonella and another on Streptococcus). This has been replaced with text and supporting citations at the start of the discussion that better describes how macrophage polarization is thought to be controlled in vivo through the changing balance of Th1 and Th2 cytokines, as well as studies that discuss the effect of Cn on macrophage NO and iNos production (page 29, lines 664-673). Additional information is also included on the known effects of Cn on signaling pathways identified in our GO data (page 31, lines 737-748). 

Minor points:

1) “In the introduction, page 3 lines 59-60, there are several more C. neoformans papers on macrophage activation and C. neoformans that have been published over the last 10+ years. The authors should reference a wider range of papers. To find these, in pubmed, search “macrophage activation Cryptococcus” or “macrophage M1 M2 Cryptococcus.””

 We thank the reviewer for suggesting this. We have added an additional five references to the existing 3. Collectively, they describe the effects of Cn infection on macrophage polarization in in vivo models and the differing fungicidal activities of M1 and M2 macrophages. These new references are as follows:

• Jain et al 2009, which shows H99 promotes a Th2 response and M2-polarization of alveolar macrophages (2).

• Muller et al 2013, showing loss of IL-4 receptor and M2 activation provides resistance to pulmonary Cn infection (3).

• Voelz et al 2009, which looks at the effect of Th1, Th2, and Th17 cytokines on the outcome of macrophage intracellular Cn infection, showing that the rate of phagosomal Cn proliferation was greater in macrophages stimulated with IL-4 than IFNγ (4).

• Hardison et al 2012 and Leopold Wager et al 2015. Both of these concern the importance of STAT1-dependent M1 polarization in macrophage antifungal activity (5,6).

2) “In the introduction, page 4, line 77 – why not reference a Cryptococcus paper? There are many papers and reviews on C. neoformans specifically available. Fungi are different than bacteria, so a fungal-specific paper would be more appropriate than a bacterial paper here.”

 We thank the reviewer for suggesting this. We have rewritten this line so that we now clearly state that Cn infection promotes a Th2-response and M2 macrophage polarization in murine models. This includes appropriate references for Cn research and review papers and the original bacterial paper citation was removed.

3) “In the introduction, page 5, line 87, the authors mention a M0-like state, but this is not defined anywhere in the manuscript. Please clarify what is meant by M0.”

 We agree that this was vague in the original text. We equate the M0 state with naïve macrophages. In our system, we essentially mean macrophages that have not previously been exposed to proinflammatory (IFNγ or LPS) or anti-inflammatory (IL-4) stimuli. The modified text reads as follows with changes underlined, “Infection with Cn also shifted the gene expression profiles of both M1 and M2 cells to a more naïve (M0) state, that is, macrophages that have not previously been exposed to proinflammatory or anti-inflammatory cytokines, while causing relatively small changes in the expression of Nos2 and Arg1.”

4) “In the methods section, page 6, line 118, the authors state that cells were given IFN-γ to “maintain the M1 polarization state.” How was this maintenance verified? This information needs to be added to the methods section.”

 We thank the reviewer for suggesting this. This has now been added to the methods section on page 6, lines 152-153. The maintenance of polarization state was tested by western blotting for iNos. As reported in our manuscript (Fig. 1B+C and 2B), iNos levels remained high 24 h post-mock or Cn-infection in IFNγ-treated samples

5) “In the results section, page 14, lines 287-290, the authors mention replacing the medium throughout the experiment at 6h intervals. This seems like quite a bit of manipulation – was this methodology compared to controls without media changes? If not, these studies need to be conducted as a control for the manipulation mentioned in the text.”

 We appreciate the reviewer’s concern about the potential impact of the medium replacement strategy used in our experiments. We completed western blots to look at the effects of intracellular Cn 24 h post-infection on iNos and Arg-1 protein levels without the use of medium replacement. These data were comparable to those included in the manuscript as figure 2B, which used the 6 h medium replacement cycles. A side-by-side comparison is included below as Figure R1.

As mentioned in the manuscript (page 15, lines 361-371), we previously conducted near-identical RNAseq experiments that did not employ medium replacement. This resulted in unexpected changes in the expression of glucose-responsive genes, making the replacement strategy an indispensable part of our approach. Additionally, all samples used in the RNAseq experiments presented in Figures 3+4 of our manuscript received 6 h medium replacement cycles, including the mock-infected samples. As both samples in these pair-wise comparisons received the same medium changes, we believe that the gene expression changes observed will most likely be a result of intracellular Cn infection rather than any manipulation.

Finally, the potential effects of the medium replacement cannot explain the upregulation of Cited1 reported in our manuscript. In western blotting experiments completed during the review process, we were able to detect Cited1 levels by 6 h post-infection in samples that were not subjected to medium replacement (page 28, lines 629-636).

6) “In the results section, page 14, line 293, the authors mention that conditions were optimized to maximize the Cn-infected M1 and M2 polarized macrophages. How was the percent infected determined? Microscopy? Flow? There is no mention of these studies in the methods, so please include these details. Similarly, this is also discussed on page 16, line 297. Again, nothing is mentioned about how the infection efficiencies were calculated. This information needs to be added to the methods section.”

 We apologize for this oversight and we have now corrected this in the manuscript by adding a description of the optimization procedure in the methods section (page 7, lines 158-170). In brief, samples infected with GFP-expressing H99S using the conditions described in the figure legend were imaged by confocal microscopy 24 h post-infection. Samples were stained with PI and calcofluor white so that dead macrophages and extracellular Cn could be identified. The percentage of live macrophages containing phagosomal Cn was determined from the recorded images. Additionally, all samples infected with Cn that were used in RNAseq assays were imaged by light microscopy immediately prior to harvest to confirm infection at the expected efficiency (page 9, lines 212-215).

7) “In the legend for Figure 4 (page 21), please add the number of experiments represented by these data.”

 Figure 4, like figure 3, presents RNA sequencing data in the form of GO analysis and STRING networks. We have now adjusted the “Analysis of RNA sequencing data” to explicitly mention the number of samples per condition harvested and analyzed for these data (page 10, lines 243-244). As this information is in the materials and methods, covering both figures 3 and 4, we did not feel the need to include this in the figure legend.

 As we felt it might have been possible that the reviewer may have meant to request the number of experiments represented in figure 5 – which did not list this in the original version of the manuscript – we have now included this together with new quantification (Fig. 5B; page 28+29, lines 640-658). While the manuscript was in review, we were able to repeat this experiment as depicted in figure 5A three times. Cited1 protein levels were higher than controls in M1-polarized Cn-infected cells in all 3 repeats. However, as the result was inconsistent for M2-polarized Cn-infected cells, where it exceeded levels in control cells in only 2 of 3 repeats, we have changed our description of this result in the manuscript to reflect this. In additional experiments that are not included in this manuscript, we observed elevated Cited1 levels as early as 6 h post-infection in M1-polarized cells in 3 of 3 repeats. For these reasons, we believe that this result is genuine and highly reproducible.

8) “In the Discussion, pages 26-27, lines 499-503, the authors reference a Salmonella paper in regards to signaling in M1/M2 polarization. It would be helpful to also include what is known about signaling from the Cryptococcus literature – there are several papers that can be referenced.”

 We thank the reviewer for suggesting this. As part of the process of making the discussion better focused on Cn-macrophage interactions (as suggested in the major comments), we have removed the Salmonella paper reference. This has been replaced with new text and references to support the notion that the primary driver of macrophage polarization during Cn infection is the broader immune response, which switches from Th2 to Th1 (Arora, S et al 2011 (7); page 29, lines 664-668). We also added additional text describing studies that provide evidence that direct interaction between Cn and macrophages can affect macrophage polarization, or at least key markers of polarization (Naslund, PK et al 1995 (8), Xiao, G et al 2008 (9); page 29, lines 668-673).

9) “In the Discussion, page 27, line 512, please indicate “cell line macrophages” and not simply “macrophages.” In addition, in the same section (lines 511-520), it would be helpful to also explain the discrepancies caused by using only a cell line instead of primary macrophages in the study.”

 We absolutely agree that this distinction should be made clear in the text and we have made the recommended change from “macrophages” to “cell line macrophages. We also agree that the potential differences in response between cell lines and primary cells should be discussed. Rather than including this as a small note within the suggested section, we added a new concluding paragraph to highlight this concern (pages 35+36, lines 872-887), including references to relevant studies, including one that directly compares the transcriptional responses of a macrophage cell line and primary bone marrow-derived macrophages to infection with an intracellular pathogen (not Cn in this case). We also indicate that we intend to repeat the work covered in the manuscript using primary cells as part of a future study. 

10) “In the Discussion, page 28, line 530, there are definitely more than 2 references for Cryptococcus studies investigating gene expression in macrophages following interaction with C. neoformans. Include the relevant studies here. Similarly, in line 538, there is more than a single study showing the activity of these signaling pathways in macrophages exposed to C. neoformans.”

 We thank the reviewer for suggesting this. In the original manuscript, we deliberately selected those studies that were most similar to ours and had used in vitro infection of cell line macrophages. We have now expanded this to include additional studies involving interactions between monocytes or macrophages and Cn. These include the following:

• Hansakon et al 2019, the authors look at how high-uptake strains of Cn affect the transcription of M1 and M2 markers in alveolar macrophages from BALB/c mice (10)

• Leopold Wager et al 2018, the authors use RNAseq-based transcriptome profiling to look at Cn induced gene expression in macrophages from mice that had been previous primed with IFN-gamma expressing H99S (11).

• Heung et al 2019, the authors use RNAseq to show that in the lung, inflammatory monocytes increase the expression of M2 macrophage-associated transcripts in response to Cn infection (12).

These studies are cited on page 30, line 716-717.

We also agree that the signaling pathways described in the line identified by the reviewer have been implicated in the macrophage response to Cn in other studies. We have added additional text describing this together with appropriate references for NF-κB (13,14), STAT1 (6,15), and TLR signaling (16-20) (Page 31, lines 737-748).

---

## [Decision Letter · Decision Letter 1]

13 Aug 2020

Intracellular *Cryptococcus neoformans* disrupts the transcriptome profile of M1- and M2-polarized host macrophages

PONE-D-20-13968R1

Dear Dr. Nelson,

We’re pleased to inform you that your manuscript has been judged scientifically suitable for publication and will be formally accepted for publication once it meets all outstanding technical requirements.

Kind regards,

Michal A Olszewski, DVM, PhD

Academic Editor

PLOS ONE

Additional Editor Comments (optional):

Reviewers' comments:

Reviewer's Responses to Questions

**Comments to the Author**

1. If the authors have adequately addressed your comments raised in a previous round of review and you feel that this manuscript is now acceptable for publication, you may indicate that here to bypass the “Comments to the Author” section, enter your conflict of interest statement in the “Confidential to Editor” section, and submit your "Accept" recommendation.

Reviewer #1: All comments have been addressed

Reviewer #2: All comments have been addressed

2. Is the manuscript technically sound, and do the data support the conclusions?

Reviewer #1: Yes

Reviewer #2: Yes

3. Has the statistical analysis been performed appropriately and rigorously? 

Reviewer #1: Yes

Reviewer #2: Yes

4. Have the authors made all data underlying the findings in their manuscript fully available?

Reviewer #1: Yes

Reviewer #2: Yes

5. Is the manuscript presented in an intelligible fashion and written in standard English?

Reviewer #1: Yes

Reviewer #2: Yes

6. Review Comments to the Author

Reviewer #1: (No Response)

Reviewer #2: (No Response)

7. PLOS authors have the option to publish the peer review history of their article (what does this mean?). If published, this will include your full peer review and any attached files.

Reviewer #1: No

Reviewer #2: No

---

## [Editor Report · Acceptance letter]

19 Aug 2020

PONE-D-20-13968R1 

Intracellular *Cryptococcus neoformans* disrupts the transcriptome profile of M1- and M2-polarized host macrophages 

Dear Dr. Nelson:

I'm pleased to inform you that your manuscript has been deemed suitable for publication in PLOS ONE. Congratulations! Your manuscript is now with our production department. 

Kind regards, 

on behalf of

Dr. Michal A Olszewski 

Academic Editor

PLOS ONE